# Environmental Resilience Technology: Sustainable Solutions Using Value-Added Analytics in a Changing World

E. Natasha Stavros [1,*], Caroline Gezon [2], Lise St. Denis [1], Virginia Iglesias [1], Christina Zapata [2], Michael Byrne [2], Laurel Cooper [2], Maxwell Cook [3], Ethan Doyle [2], Jilmarie Stephens [1], Mario Tapia [2], Ty Tuff [1], Evan Thomas [4], S. J. Maxted [2], Rana Sen [2] and Jennifer K. Balch [1,5]

[1] Earth Lab, Cooperative Institute for Research in Environmental Studies (CIRES), University of Colorado Boulder, Boulder, CO 80309, USA; lise.st.denis@colorado.edu (L.S.D.); virginia.iglesias@colorado.edu (V.I.); jilmarie.stephens@colorado.edu (J.S.); ty.tuff@colorado.edu (T.T.); jennifer.balch@colorado.edu (J.K.B.)
[2] Deloitte Consulting, LLC, New York, NY 10112, USA; cgezon@deloitte.com (C.G.); chzapata@deloitte.com (C.Z.); michbyrne@deloitte.com (M.B.); lacooper@deloitte.com (L.C.); martapia@deloitte.com (M.T.); smaxted@deloitte.com (S.J.M.); rsen@deloitte.com (R.S.)
[3] Department of Geography, University of Colorado Boulder, Boulder, CO 80309, USA; maxwell.cook@colorado.edu
[4] Mortenson Center in Global Engineering and Resilience, University of Colorado Boulder, Boulder, CO 80309, USA; ethomas@colorado.edu
[5] The Environmental Data Science Innovation & Inclusion Lab (ESIIL), University of Colorado Boulder, Boulder, CO 80309, USA
[*] Correspondence: natasha.stavros@colorado.edu; Tel.: +1-(858)-254-5939

**Featured Application: We provide a case study for the Wildfire Response, Risk Mitigation and Recovery and a methodology for research-to-commercialization (R2C) for analytics of value using solutions-oriented science.**

**Abstract:** Global climate change and associated environmental extremes present a pressing need to understand and predict social–environmental impacts while identifying opportunities for mitigation and adaptation. In support of informing a more resilient future, emerging data analytics technologies can leverage the growing availability of Earth observations from diverse data sources ranging from satellites to sensors to social media. Yet, there remains a need to transition from research for knowledge gain to sustained operational deployment. In this paper, we present a research-to-commercialization (R2C) model and conduct a case study using it to address the wicked wildfire problem through an industry–university partnership. We systematically evaluated 39 different user stories across eight user personas and identified information gaps in public perception and dynamic risk. We discuss utility and challenges in deploying such a model as well as the relevance of the findings from this use case. We find that research-to-commercialization is non-trivial and that academic–industry partnerships can facilitate this process provided there is a clear delineation of (i) intellectual property rights; (ii) technical deliverables that help overcome cultural differences in working styles and reward systems; and (iii) a method to both satisfy open science and protect proprietary information and strategy. The R2C model presented provides a basis for directing solutions-oriented science in support of value-added analytics that can inform a more resilient future.

**Keywords:** innovation; commercialization; decision making; human-centered design; information technology; data analytics; resilience; environment

## 1. Introduction

We are now in the Anthropocene—an unprecedented time of global environmental change caused by urbanization [1], desertification [2], biodiversity loss [3], and more frequent extreme events [4]. The drivers for this change include global warming [4] and

rapid population growth [5]. These environmental changes are likely to threaten people's lives, livelihoods and assets in years to come [2], motivating a need to think about the mitigation of impact.

The mitigation of impact is broadly referred to as environmental resilience—the prevention, ability to withstand, respond and recover from environmental perturbations and shocks [6]. Some leading questions in disturbance and resilience ecology include the following: (1) How and why does life organize across scales? [7]; (2) How will disturbance impact how life organizes across scales? [8]; and (3) How do we mitigate our climate risk to ecosystems and communities as a function of vulnerability, exposure and hazard? [9]. Researchers are now well-positioned to examine these questions and inform a more resilient future as we navigate the information age with cutting-edge analytics and big data on Earth observations from diverse data sources such as remote sensing [10] and social media [11].

It is increasingly evident that new technologies informed by climate and environmental research, as well as new applications of existing technologies, are necessary to support resilient community mitigation and adaptation from environmental change (Figure 1). Such technologies may include remote sensing, machine learning, artificial intelligence, geospatial analytics, sensors, and social media data that can be synthesized and used to support decisions in the natural, built and social environment; it may also include technologies that can directly increase resilience, such as improved built infrastructure, water management, and renewable energy generation. *We define environmental resilience technologies as data-driven, **value-added analytics** informed by **solutions-oriented science** that enable society to become more resilient to changing environmental futures.* Within this definition of environmental resilience technology, we further define value-added analytics and solutions-oriented science as follows.

At present, there are dozens of solutions for different applications, yet one of the biggest limitations is sustaining the applications for use through time [12]. As such, innovation requires not merely creating a piece of technology but also developing the business model for sustained use. Because value inherently has cost [13], in the context of solutions using analytics, *value-added analytics* are worth paying for and have the ability to go from research to commercialization.

A methodology for commercializing a product or service is human-centered design (HCD) [14,15]. HCD is a bottom–up approach to understand people, their actions, decisions, and feelings. It is rooted in anthropology and engineering [16–18]. HCD starts from the study of people in a problem or opportunity space and leads to the generation of solutions from what is learned. HCD focuses on human desirability and includes the human perspective in all steps of problem solving. Some limitations of HCD include incremental design changes based on the human-centric perspective, typically within a single application rather than broader examination of related systems and the decision space [19]. This often occurs when designers focus on what people directly ask for rather than the root causes behind those desires [20,21]; this is especially true in data science applications [22].

Alternatively, the WKID (pronounced wicked) Innovation framework focuses on transparent and systematic traceability across the decision space to identify information technology requirements and inform data science in support of closing information gaps. WKID Innovation tackles wicked problems [23], which are known to have no single cause or best solution. In modern society, multiple organizations often tackle wicked problems, presenting differences in values across entities that complicate decision making. Decision making uses two elements [24]: (1) data, which can be quantitative or qualitative, and (2) values, which are represented as the subjective weighting of costs and benefits of the outcome of that decision. While data may provide guidance for potential outcomes, often, the solution depends on different users' values, which often results in disagreement about the cause of and solution to the problem. WKID Innovation [25] uses the Knowledge Hierarchy [26], the PEST (Policy, Economics, Sociocultural Factors, and Technology) model [27], NASA system engineering [28], Theory of Change [29], and to direct scientific

discovery that informs how people and communities act, which is henceforth defined as *solutions-oriented science*.

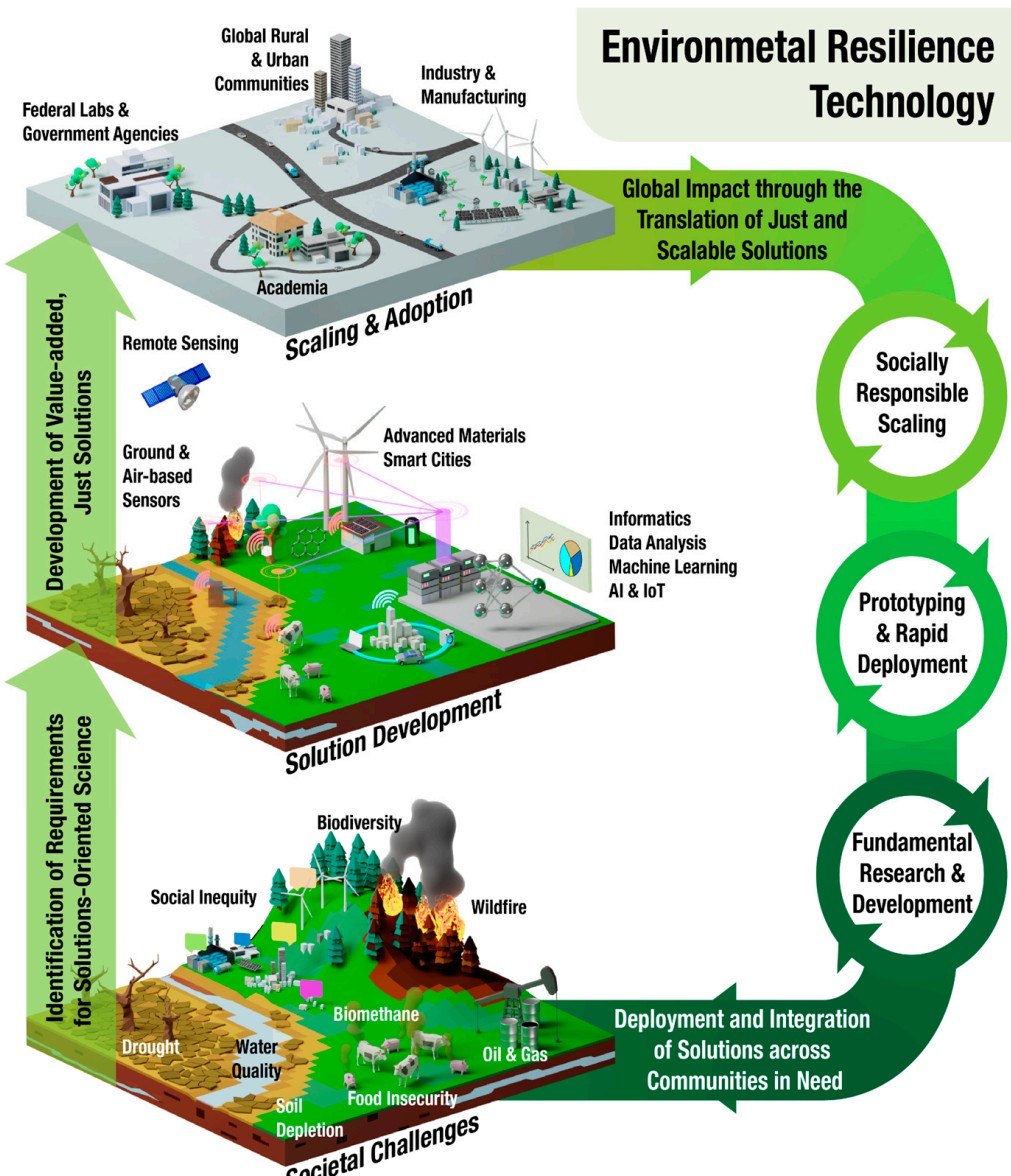

**Figure 1.** Environmental Resilience Technology addresses societal challenges through solution development that involves cross-sector collaborations for scaling and adoptions. Specific to this manuscript, we develop a model for solutions-oriented science that provides value-added analytics.

Transitioning research to commercialization can be achieved through academic–private partnerships. Establishing effective academic–private partnerships, however, is challenging. In this paper, we provide a research-to-commercialization (R2C) model to support academic–private partnerships that develop environmental resilience technology. The R2C model aims to achieve the following: (1) systematically identify information gaps in the decision space, (2) define algorithm requirements, and (3) develop sustainable solutions for resilience to global environmental change through market viability. The model is based on a pilot project of the Deloitte Consulting and the University of Colorado Boulder (CU Boulder) Climate Innovation Collaboratory (CIC). This paper describes the outcomes achieved from the wildfire analytics project to support community wildfire preparedness, resilience, and

response. In the following sections, we apply our R2C model to a case study to address the wicked wildfire problem in the United States.

*Case Study Background: The Wicked Wildfire Problem in the United States*

For the scope of this work, we explored how data analytics could support megafire mitigation, resilience, and response. The wicked wildfire problem refers to the fires that matter or the fires that have negative social impact [30]; these are commonly referred to by the public as "megafires". Megafire is a sociopolitical term and therefore has no standard unit, but in the western United States, a megafire generally refers to fires that meet some or all of the following criteria: (1) they burn more than 50,000 acres or ~20,234 hectares [26], (2) they produce smoke affecting millions of people over regional scales [31], (3) they threaten residential homes or community structures [32,33], and (4) they may be fast moving, creating entrapment scenarios for local residence and making suppression efforts even more difficult.

Megafires result from the interactions between flammable fuels, climate, and at least one source of ignition [34]. These interactions do not share a common mechanism, so managing for megafires requires a complex understanding of multiple systems and the ability to execute complex decision making across those systems. Climate change from greenhouse gasses emitted from the burning of fossil fuels [4] results in warmer, dryer, and windier conditions that are ripe for increased fire danger [35] and the likelihood of large fires in the continental United States [36–38]. Despite fuel constraints from increased fire in the future, the fire area is expected to increase [39]. Ignition rates for fires, including from lightning and people (e.g., prescribed fire, arson, overheated cars, power lines, bonfires, fireworks, etc.), are variable. Humans change the fire season, extending it throughout the year and sometimes peaking outside when lightning ignitions are possible [32]. Furthermore, humans change where fires occur and how proximal they are to settlements [33] through urban expansion into wildlands. Finally, the United States has experienced nearly a century of fire exclusion (putting out all fires) and limited prescribed burning that has led to extreme fuel accumulation [40], affecting fire behavior substantially in some regions [41]. Fuel accumulation not only affects fire behavior, it can also affect the condition of the fuels as vegetation competes for resources and becomes stressed, worsening burn severity [42]. Land use and management has also contributed to fire frequency and severity as well as the type of fuels available to burn. For example, land use influences the presence of human development and its influence on homes as fuel [33,43,44] as well as the presence of invasive species [45,46] that affect fire occurrence and frequency [47]. Putting all these ingredients together, megafires have the highest risk to society [30], and many actors play an important role in mitigating their impact.

Because different actors are responsible for different contributors to the wicked wildfire problem, there are no single solutions. Furthermore, there are a plethora of technologies aimed and designed for different actors across the disaster lifecycle from pre-fire resilience, preparedness, and hazard mitigation to active fire detection, tracking, and response to post-fire recovery [12]; Supplemental Material S1. In fact, a lack of technology is not the biggest challenge facing the wicked wildfire community; instead, the following challenges apply: (1) a need for strategic and coordinated efforts, (2) access to data and standardization, (3) research and development that thinks holistically about the problem across the disaster life-cycle in the context of resilience; and (4) considerations of financing solutions for long-term sustainability [12,48]. Specific to point four, the wicked wildfire problem provides a strong use case for testing the R2C model.

## 2. Methods

### 2.1. Research-to-Commercialization (R2C) Model

We developed a research-to-commercialization (R2C) model (Figure 2) that integrates the WKID Innovation and HCD methodologies. The HCD approach engages a variety of tools and frameworks to assess user needs and market conditions. Leveraging the

outcomes of interviews, we developed user stories, use cases and an ecosystem map to define the current market landscape. To find the intersection of innovation potential, we estimated the user desirability, business viability, and technical feasibility of a development based on qualitative and market research. Our HCD process brought potential end users into the design process through interviews and focus groups to define the problem set, narrow potential solution sets, and test initial solution prototypes and wireframes. The HCD process also aims to ensure end user needs are met and aligned to organizational requirements by reviewing enterprise architectures for industry-provided technologies capable of scaling to support big organizations.

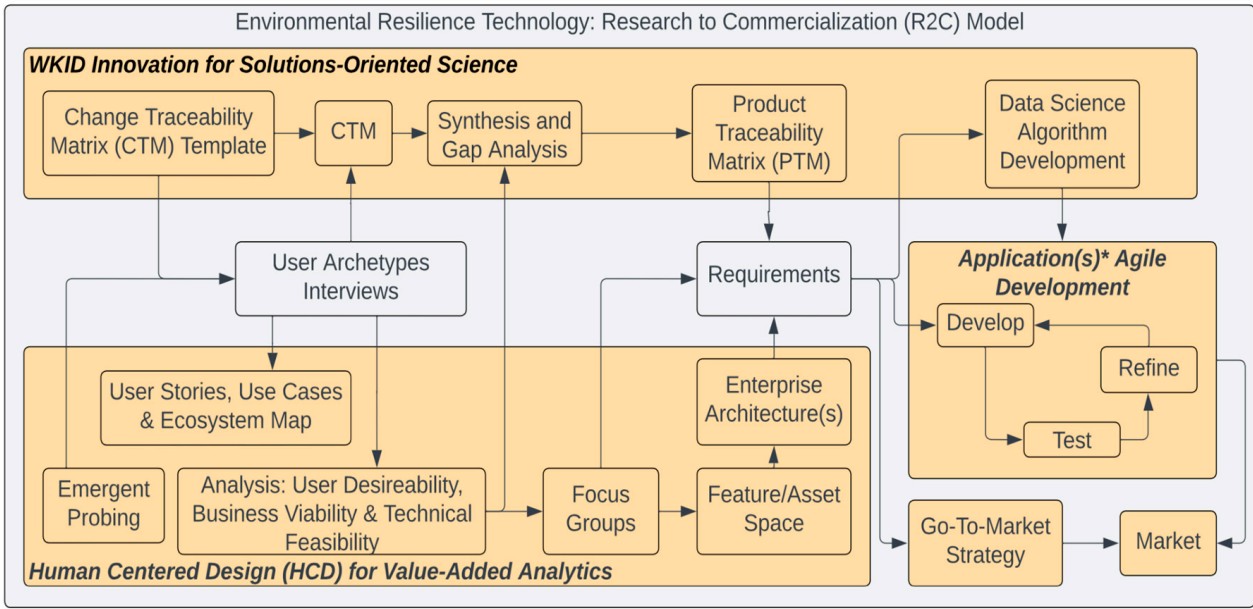

**Figure 2.** An environmental resilience technology research-to-commercialization (R2C) model can identify solutions-oriented science for value-added analytics that leverage global change research to inform decisions around a more resilient future.

WKID Innovation uses two core tools: the Change Traceability Matrix (CTM) and a Product Traceability Matrix (PTM) [25]. The CTM organizes information to provide a comprehensive look across the decision space to identify common information needs and specifications to change the way decisions are made. It documents everything from key performance indicators (KPIs, e.g., "# lives saved" or "$ saved"), the decisions being made, by whom and for what purpose, how decisions are funded, what motivates the decision, the tools currently in use, the information needs, gaps and limitations, as well as key parties engaged and affected by the problem. The PTM provides high-level "requirements" on product definition. WKID Innovation does not assess market viability explicitly but rather assumes that if there is a need and it fits the decision space, it has value, which may not always be the case.

### 2.2. Case Study: Dataset Curation

In the R2C model, the WKID Innovation framework provided the basis to structure the initial interviews rather than the more emergent probing used in HCD for initial user interviews. We conducted 26 interviews across four gradients of representative decision makers, creating user archetypes (Figure 3: Resilience, Public Information Officer, Land Planning, Recovery, Land Managers, Policy Implementation/Enforcement, and Emergency Management). The first gradient spanned decision makers in pre-fire, active fire, and post-fire situations such as emergency responders at local, state, and federal levels; public information officers (e.g., communications manager); land use and asset planners (e.g.,

developers, real estate, (re-)insurance companies, city planners, and utility companies); recovery (e.g., public assistance, emergency management organizations, and (re-)insurance companies); resilience and preparedness planning (e.g., local, state, and federal offices), land managers (i.e., private, state, federal, and tribal partners), and policy implementation and enforcement agents. The second gradient included people working in public, private, and non-profit sectors. The third gradient spanned decision makers from rural to urban environments. Finally, we interviewed across the gradient of decision maker budgets from low to high. User personas were created based on the sector (public, private, non-profit, academic), the domain of jurisdiction (local, state, regional, national), and the user archetypes.

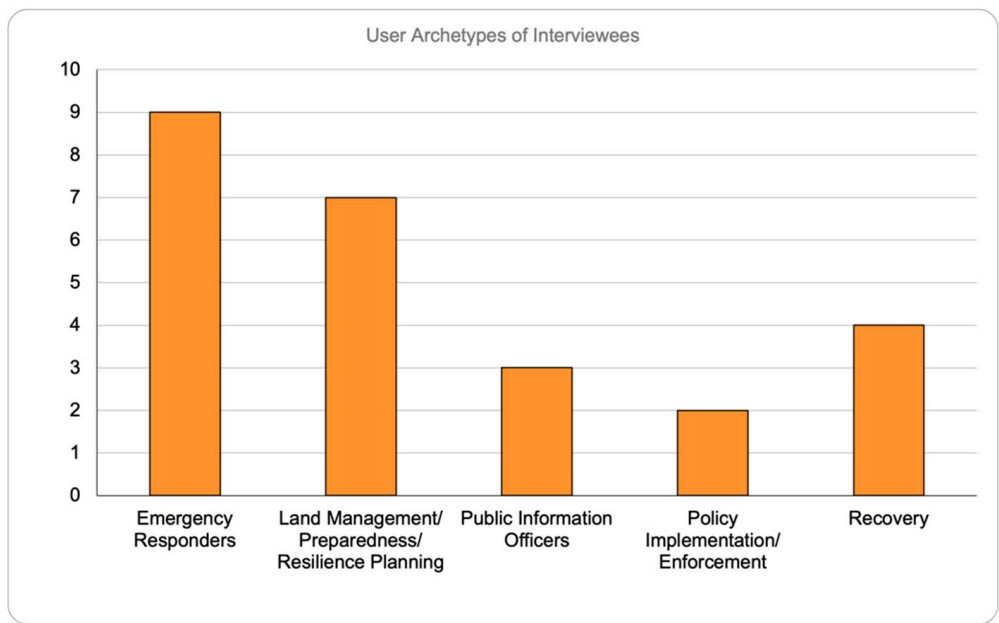

**Figure 3.** Number of interviewees by user persona. Note that here, we show five personas when the analysis determined 8 personas because Emergency Responders, Land Management/Preparedness/Resilience Planning, and Public Information Officers were further partitioned into local and state/regional personas.

*2.3. Case Study: Information Gap Analysis*

For the interviews, we standardized questions (Supplementary Materials S2) and used a standardized online data collection form to take notes as a precaution against subjective interviewer bias. For each interview, there was a lead interviewer and two note takers, one taking verbatim notes and the other synthesizing key points emphasized by the interviewee. The questions (Supplementary Materials S2) paralleled the structure to populate a Change Traceability Matrix (Supplemental Material S1). The online data collection form generated a shared spreadsheet with 36 columns: A through AJ. These columns were then cross-walked to the CTM to show where key information from explicit questions likely populate into the CTM (Figure 4).

To populate the CTM, we then pulled key information from responses into synthesized statements relevant to each column of the CTM. We created a separate CTM per User Archetype. Each interview response was parsed into the appropriate User Archetype by creating a new row for each KPI starting left to right. Sometimes, one decision would have multiple KPIs as goals or objectives; in these cases, the decision descriptor was a merged cell spanning all associated KPIs. Similarly, any one KPI may rely on multiple types of information. A new row was added for each piece of needed information, and the KPI cell related to that decision was merged to span all needed information. While every effort was made to ask all questions, sometimes, interviewees answered questions not yet asked in

response to earlier questions, and sometimes, interviewees simply did not understand or could not answer a question. In these cases, any questions for which participants did not provide details were left blank in the final CTM and may represent gaps in information from the interviews or and gaps in the system. Bolded text in the CTM represents key quotes from interviews that capture a recurring sentiment by interviewees within that persona. Both the verbatim and synthesis notes (each taken by a different note taker) were treated independently (as different data points) and analyzed through a double-blind process whereby the interviewee and the analyzer were not identified to either party. As information from the interview overlapped with existing synthesized information in the CTM, no new rows were created, but existing rows were edited to provide more clarity. All details specific to a company, agency, or organization were removed for anonymity. The CTM provides a synthesis capturing general methods, mechanisms, and patterns for a user persona and each key decision they make.

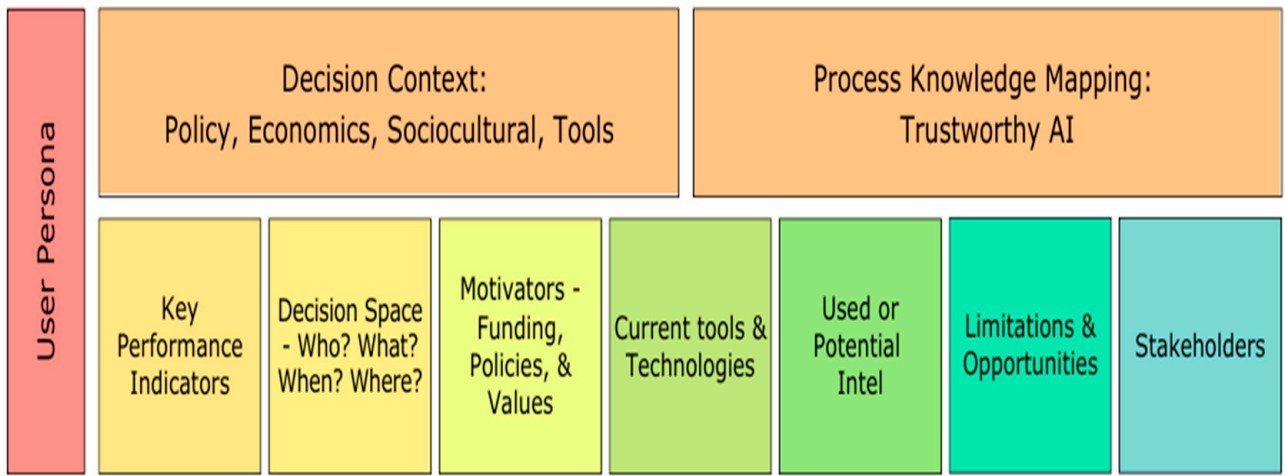

**Figure 4.** A cartoon of the traceability for organizing information from interviews into the Change Traceability Matrix (CTM).

Prior to interview synthesis, the CU Boulder researchers presented nine data analytics capabilities in development, which were used as potential solutions for further refinement. As part of the HCD process, the joint Deloitte–CU Boulder team spent several days qualitatively synthesizing the interviews, documenting individual pain-points and solution areas with sticky notes, and aggregating information into cross-cutting insights and themes. These insights were later validated through the CTM analysis.

We then used HCD to gauge human desirability, business viability, and technical feasibility. We applied a desirability–viability–feasibility, or "DVF" framework [49], to the nine CU Boulder researcher data analytic capabilities, and we down-selected three that scored the highest. To assign scores, the integrated CU Boulder–Deloitte team discussed research insights to collaboratively arrive at desirability scores. The Deloitte team led the scoring for business viability, using market research and subject matter expertise from project advisors working in the field. The CU Boulder team led the scoring for technical feasibility, using domain expertise for feasibility to develop. The initial areas selected for further research and development included several data analytics for "dynamic risk", "evacuations" and "public perception".

Using the identified value-added analytics, a PTM (Supplemental Material S1) was created. We used the CTM and looked across all user personas, identifying any time that "risk" (and "hazard", "exposure", or "vulnerability" as these are inclusive of the definition of risk [30]), "evacuation", and "public perception" (or "social media" or "# likes and retweets") were identified as needed information (Figure 4, "Limitation or Opportunity column). Each information need was imported as a row into the PTM and categorized: Product Definition; User Experience ("UX")/Training & Personnel; Accessibility; Business

Model; Trustworthy AI [50], Product Requirements; Other Tech & Culture Needs; Information Gaps. PTM rows were grouped based on similarity, and a requirement was written. Each requirement was cross-referenced with the number of rows and unique personas they represented. The number of user personas was recorded.

Two analyses were performed on the PTM recording: (1) the number of user personas, and (2) the number of decisions relying on the highest priority, market-viable information: "dynamic risk" and "public perception". The number of user personas and user stories were determined based on information needs specifically using the following terms: "public perception", "dynamic risk", "vulnerability", "hazard", "exposure", "filtered communications", "evacuation", "misinformation", "# of likes and retweets". One user persona may make many different decisions for different reasons. Decisions were recorded as user stories. User stories followed this format: To make a [decision] for [purpose], a [user persona possibly including generalized job descriptor, e.g., "local emergency response firefighter"] relies on [information]. Because each interviewee provided a job descriptor and was assigned a persona, stories could be verified with interviewees while maintaining the anonymity of respondents. In total, there were 39 user stories from 8 user personas.

### 2.4. Case Study: Proof-of-Concept Demonstrations

Following the HCD process and using the PTM analysis of user stories to further define each information variable, we developed three proof-of-concept analyses on a single megafire event, the 2021 Colorado Marshall Fire. Here, we describe the methods used to create proof-of-concept analyses, which were used in continued iterations of qualitative research to gather feedback from potential users.

The Colorado Marshall Fire occurred on New Year's Eve (30 December 2021 to 3 January 2022) in a suburban neighborhood in the central United States with a traditionally low fire hazard. Tens of thousands of people were evacuated and over a thousand homes burned. The Marshall fire occurred in the wildland urban interface (WUI) with human development adjacent to a large expanse of naturally preserved open space. The transport network is dominated by slow speed, curved residential streets. As the fire traveled through the WUI with gusts greater than 100 miles per hour [51], evacuees flooded onto small, circuitous streets and sat in traffic trying to get away from the fire blowing overhead.

Because of recent investments in climate resilient infrastructure and the need for people to evacuate away from fire hazard, we applied analytics for the following: (1) social media filtering, (2) evacuation rate predictions, and (3) predicting the annual fire hazard from present to 2070. Social media filtering provides context for the chronological order of events and public perception of the fire as it was happening. We use social media data collected using the Twitter API version 2 [52] to extract all original tweets (no retweets) and containing the hashtags *marshallfire* or the words "*Marshall Fire*". Tweets were collected from 30 December 2021 through the end of January. We collected beyond the containment date to capture the post-fire discussion related to the evacuation and returns. We pulled 26,788 tweets, 1756 of which related in some way to the evacuation. These 1756 tweets were further stratified using a neural net classifier to identify the tweet contributor role [11,53,54]. The content of these 1756 tweets was examined manually, including the related conversational threads and linked sources, to build a timeline of events We filtered media sources and focused on official messaging and contributions from people directly impacted by the event. We identified the key issues and communication disconnects related to the evacuation as it evolved and the socio-technical innovations supporting decision making and communication. Based on our initial analysis, we expanded the data collection to include the names of the towns under evacuation orders (Louisville, Superior, Boulder, and Lafayette) and tweets related to evacuation or traffic. For example, we searched for original tweets for the town of Louisville using terms *Louisville* plus either the word *traffic* or *evacuation* during the evacuation timeframe. After eliminating noise (e.g., tweets reporting

*traffic* in *Louisville*, KY or marketing tweets promoting *superior* internet *traffic*), this resulted in the addition of 232 tweets.

Simulations of evacuation traffic [55] away from the Marshall fire used three inputs: a network representation of the roads provided by Open Street Map, a fire perimeter from the National Interagency Fire Center [56] showing the extent of the burned area, and a list of origins and destinations for travelers evacuating the fire. We made a naive estimation of origins to be one vehicle leaving each destroyed structure. Structure locations came from the Boulder County Sheriff's office and were geocoded to match building records from the Zillow ZTRAX dataset [57]. We estimated isoclines within different amounts of travel time.

We used existing research to predict future fires from 2020 to 2060 in the contiguous U.S. [58]. We overlaid major road networks to better see where infrastructure exists, and hazards were changing.

Deloitte led the agile HCD process refining enterprise technologies for usability, and the CU Boulder team adapted algorithms to provide value-added analytics in the spatial and temporal resolutions relevant for decision making.

## 3. Results

### 3.1. Case Study: Information Gap Analysis Results

Our analysis of the CTM (Supplemental Material S1) showed that the most highly needed information across user personas was related to Public Perception and Dynamic Risk (Figure 5a). The most needed information across user stories, however, was Public Perception, as it relates to how the public perceives fire risk (Figure 5b) as fire hazard (i.e., the potential of fire to occur). Information on fire hazard was equally as important (Figure 5b) as information on exposure (Figure 5b), while information on how vulnerable a community is, was more needed across the decision space (Figure 5b).

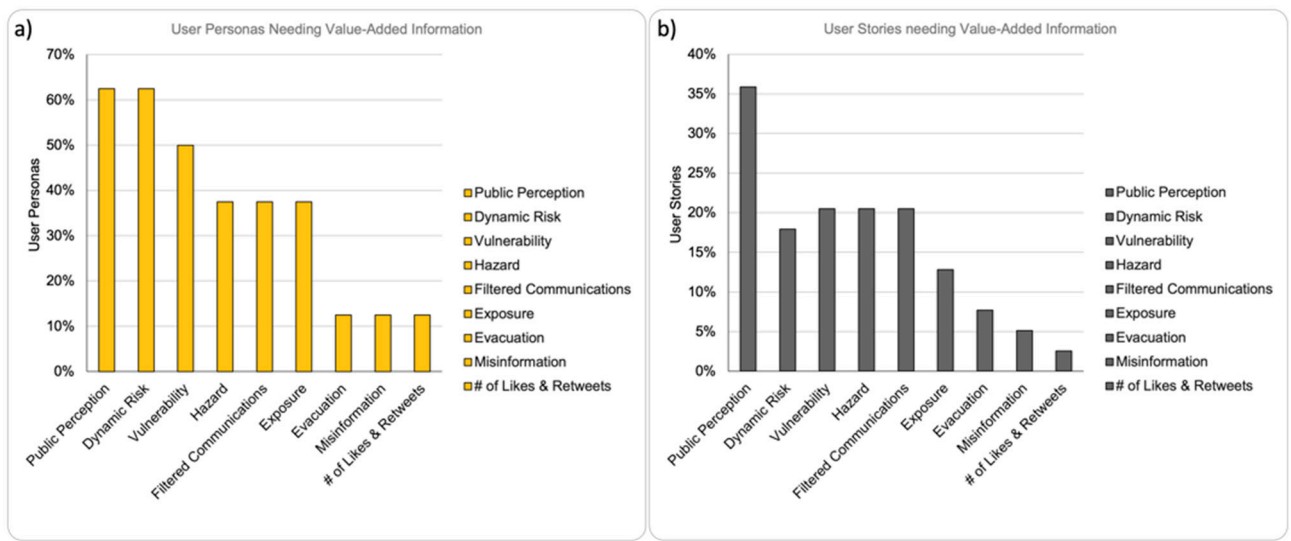

**Figure 5.** Synthesis of the Change Traceability Matrix (CTM) focusing on (**a**) percent of user personas (out of 8) needing information directly related to or relying on filtered social media or evacuations from high fire risk areas defined as a function of hazard, exposure, and vulnerability; and (**b**) percent of user stories (out of 39) relying on this information to make a decision.

Our analysis showed that solely relying on HCD would have excluded the development of social media filtering, despite being the most needed piece of information across user personas and stories (Figure 5). While HCD identified some of the root problems, information needs for megafire mitigation and the creation of more effective solutions, it did not capture the unimagined solution of social media filtering. Specifically, because few technologies exist (if any at all) to filter social media data and there is often a widespread mistrust of information shared on social media, it was hard for interviewees to imagine

a solution for using social media as a basis to inform public perception. By questioning interviewees about what drives decisions and the missing information, the systematic analysis from WKID Innovation highlighted the overarching need for information on public perception.

Despite hazard information being most valuable across decisions, the most common information gaps across personas (Table 1) are in social media filtering and risk futures. Recognizing the need for not merely filtering by hashtag, it is critical to use of artificial intelligence for identifying deep fakes, misinformation, bots, verified accounts or unique contributions to the conversation during natural disasters. Risk futures include consideration for how we message risk and not just how we have historically calculated it as acres burned.

**Table 1.** Information gaps identified across user personas and a summary definition listed as a requirement for providing that information.

| Type of Limitation or Opportunity | # of User Personas | Requirement Consolidated Across Personas |
|---|---|---|
| Social Media | 5 | Social media information shall include filters by: "deep fakes", misinformation, bots, verified accounts, etc. |
| Risk Futures | 5 | Risk futures that project risk, as defined by how it is messaged rather than just acres burned, under different management scenarios to link cost of management to risk mitigation benefit. |
| Risk General | 3 | Risk information shall provide uncertainty by each layer: hazard, exposure, vulnerability. |
| | | Risk information should consider scalability beyond data limitations of the United States. |
| | | Risk information shall include more than simple maps of the Wildland Urban Interface. |
| Hazard | 2 | Hazard information shall provide fuel maps that are updated frequently as fuels change. |
| Vulnerability | 2 | Vulnerability information shall include building ignition potential today and into the future. |
| Incident Reporting | 2 | Incident information shall automatically populate based on curated data from different data sources. |
| Exposure | 1 | Exposure information shall include building locations today and likely locations into the future. |
| General | 1 | Information technologies shall focus on proactive solutions rather than only reactive solutions (i.e., suppression). |
| | 1 | Information of value shall include metadata. |
| | 1 | Impact information shall link building damage to insurance policies. |
| | 1 | Information of value shall be verified with local knowledge. |
| | 1 | Information of value shall provide the granularity needed to inform decisions. |
| | 1 | Information technologies shall enable analytics (e.g., trend analyses). |

The PTM showed that the most valuable requirement for any information technology across user personas (Table 2) was the need for interoperable "plug-in" technologies that work with decision makers' existing tools. Next was the need for information technologies to be accessible via limited connectivity and with limited compute resources. Finally, all information should be intuitive to interpret, and information should be consistent when scaled between federal reporting to local decision support for implementation.

**Table 2.** This is the Product Traceability Matrix that provides requirements for providing information and information technologies of value.

| Type of Limitation or Opportunity | # of User Personas | Requirement Consolidated Across Personas |
|---|---|---|
| Product Definition | 7 | Information technology shall be interoperable to "plug in" to existing data portals used by User Personas to reduce the number of sources/screens that they must visit and enable them to use existing data layers. |
| | 1 | Data platforms should plug into a single existing government data portal when one becomes available by the federal government. |
| User Experience (UX) | 5 | Information layers shall be intuitive to interpret to reduce training for use. |
| Accessibility | 6 | Information technologies shall be accessible via a cell phone or government laptop with limited connectivity. |
| | 1 | Information layers shall be accessible via both information technologies and print outs. |
| | 1 | Information layers shall be archivable with provenance to be public record. |
| Business Model | 5 | Information technologies shall meet the objectives of federal funding sources while also servicing local and state decision needs. |
| Trustworthy AI | 4 | Information for value-added analytics shall have transparent documentation of algorithms. |
| | | Information for value-added analytics shall be open source. |
| | | Information for value-added analytics shall include uncertainty and error propagation. |
| Product Requirements | 5 | Information for value-added analytics shall be archived for long-term access. |
| | | Information for value-added analytics shall be pre-processed and ready to use. |
| | | Information for value-added analytics shall incentivize more resilient behavior and penalize less resilient behavior. |
| | | Information technologies shall integrate cybersecurity. |
| | | Information technologies should be marketed to the relevant agencies for using the available information. |
| Other Technology and Cultural Needs | 1 | Information technologies should include a business model to better serve less advantaged communities without exploiting them. |

*3.2. Case Study: Proof-of-Concept Demonstration*

As part of the HCD process, we provided some analytics of value for determining evacuations from a fire (Figure 6) and how public perception influences our ability to communicate and keep communities safe (Figure 7). The first evacuation notice appeared on Twitter at 12:57 pm but does not include specifics about which areas were affected by the evacuation order. Official guidelines specified that if you see flames, evacuate. Shortly after, individual replies provided details about which areas were under evacuation including the entire town of Superior. Over the next hour, video was shared by those returning to grab their belongings and pets that documented traffic flow into the cities of Superior and Boulder, the first areas under mandatory evacuation. Within an hour, traffic was at a standstill for both Superior and Boulder. Shortly after 2 pm, the town of Louisville was ordered to evacuate. There was a steady stream of communications as residents evacuated and shared updates from their mobile phones while stuck in traffic. Information was shared about unforeseen events such as traffic light outages and freight trains, also disrupted by the fire, that were blocking surface streets. By analyzing evacuation potential (Figure 6) at walking speed, we see that it can take hours to escape the fire perimeter and surrounding areas. Pedestrians or disabled travelers without access to motorized transport may not be able to cover enough ground to find a safe destination. These would not be concerns if

traffic were traveling at normal speeds because residents would have the entire Denver Metro Area to find friends, family, hotels, or shelters.

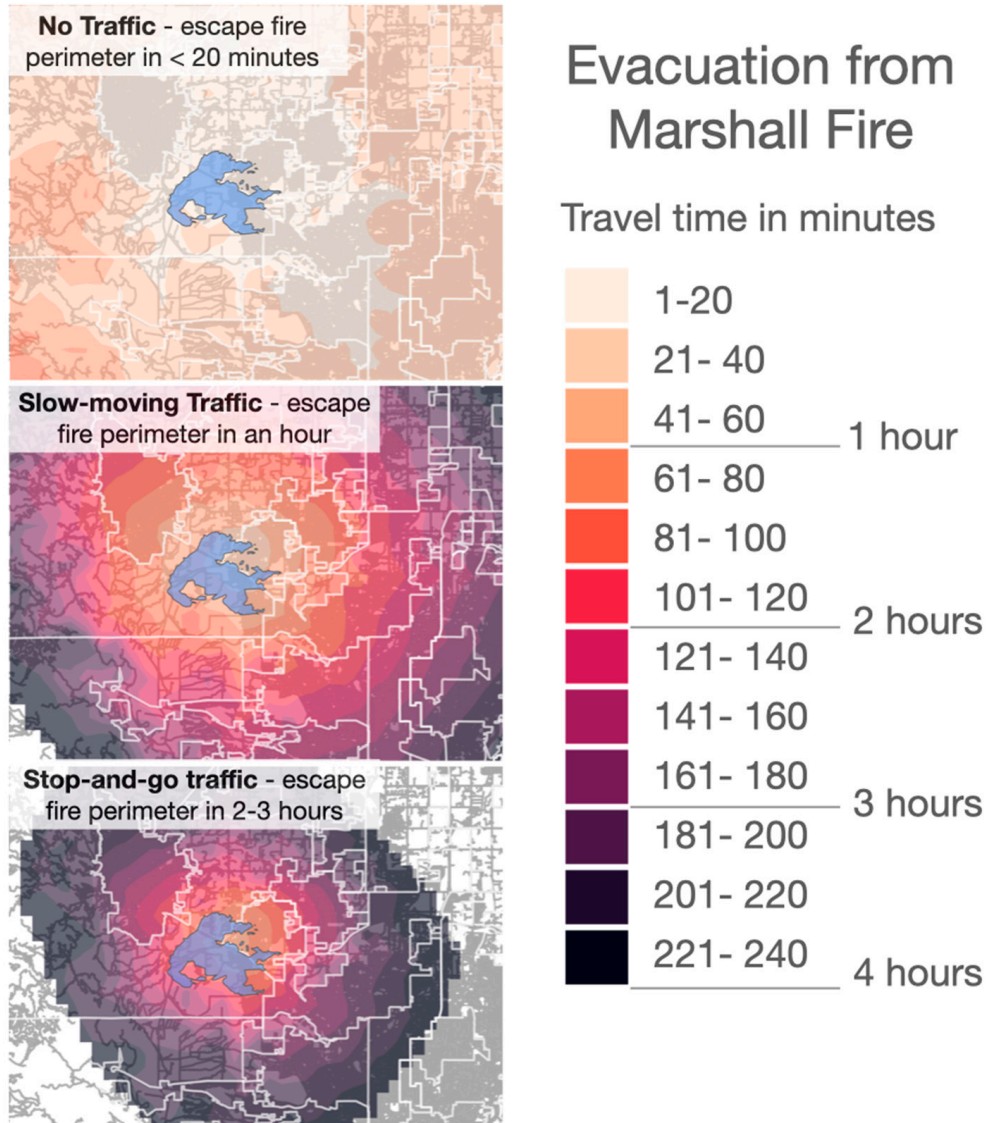

**Figure 6.** Better understanding evacuations: Utilizing data analytics to advance understanding of bottlenecks during natural disasters such as during the Marshall fire in Boulder County in December of 2021. The blue polygon is the Marshall fire perimeter. White lines are municipal boundaries within the evacuation zone. Gray lines represent the transport network. The first panel shows the normal traffic conditions where a driver can easily escape the fire perimeter within 20 min. The second panel shows moderate traffic conditions where the average travel speed is 8 miles per hour. The third panel shows heavy traffic conditions moving at an average of 4 miles per hour. Four miles per hour is approximately walking speed, showing that there are locations where it would be difficult to escape the fire.

Results from the social media filter juxtaposed with the evacuation maps highlight the role the public plays in bridging information gaps between official reporting and Geographic Information System (GIS). Tweets from evacuees provide destinations and public perceptions about evacuation messaging that can be used in modeling. Furthermore, more efficient evacuation routes can be found using machine learning of observed traffic patterns from cameras, real-time navigational tools, and personal tweets. Furthermore,

tweets from official sources can highlight key breakdowns in communication and the lack of cross-entity coordination during the evacuation.

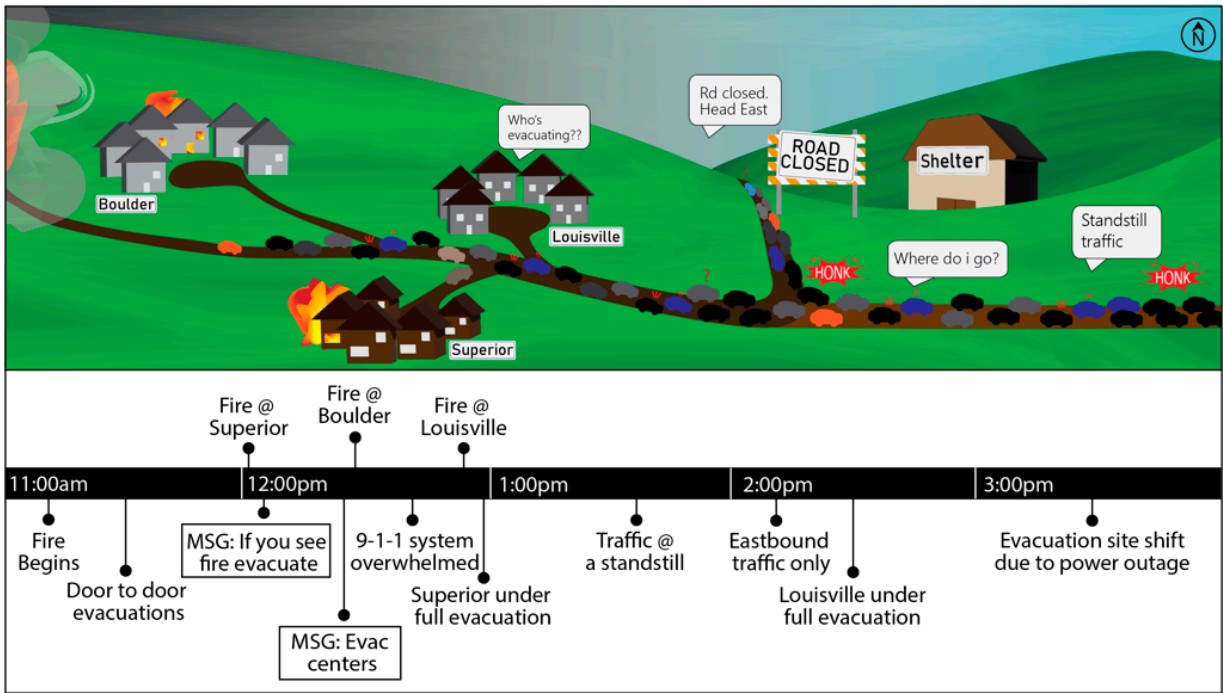

**Figure 7.** Twitter in a disaster: Information content from social media can provide critical context to understand evacuation patterns during natural disasters, such as the Marshall fire in Boulder County in 2021.

The results of our analysis that overlaid major road networks with increased fire hazard, characterized by larger and more frequent fires in the future as a result of climate change [36], shows that more places in the southeastern, midwestern and northeastern United States are likely to experience increased burden on infrastructure for evacuations (Figure 8).

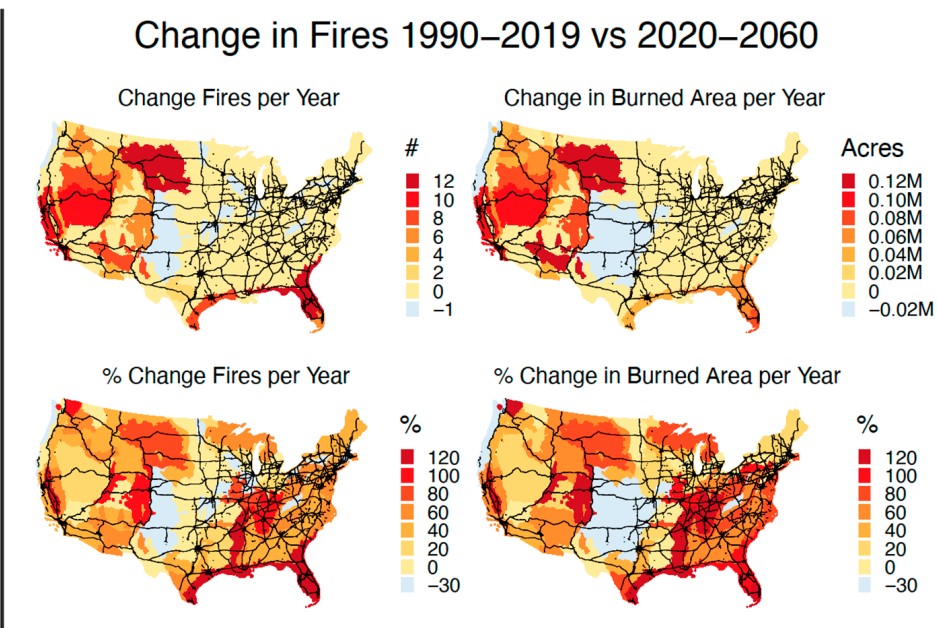

**Figure 8.** Climate-driven changes in future fire: We show increases in fires per year and burned area and as percent change between 1990–2019 and 2020–2060 with current major USA road networks to identify where roads are in the context of increasing fire hazard.

## 4. Discussion

### 4.1. R2C Model for Co-Production of Sustainable Solutions in Environmental Resilience Technology

Environmental resilience technology offers great promise for shifting science from observing environmental change toward providing solutions that serve society. An important step is creating *analytics of value* using, for example, machine learning and artificial intelligence [59]. Moreover, cutting-edge artificial intelligence and data fusion can lead to important and robust predictions about our possible environmental future [60,61]. There is a growing call to leverage science to better inform market solutions [10,62].

The R2C model, demonstrated in this paper, reflects the integration of public–private capabilities [48,63]. The R2C model integrates the WKID Innovation framework [25] with HCD, maps the information gaps across multiple user types and identifies the greatest gain while developing market-driven solutions. WKID Innovation provides an in-depth systematic analysis of the decision space to inform *solutions-oriented science*. Research on its own, however, does not precipitate sustainable solutions. Sustainable solutions consider a funding model beyond preliminary research or development [12]. Designing and clearly stating the requirements for how research fits into the decision space at the outset ensures an off-ramp from research to application [64] but does not mitigate risks for sustaining operations [65]. By integrating WKID Innovation with HCD and market research, we leverage the strengths of each framework to mitigate risks to technology adoption. Arguably, R2C offers a model to scale use-inspired translational research by environmental science domain experts [66] and drive tomorrow's technologies and solutions.

R2C sits in the research-to-commercialization taxonomy of "contract research and consultancy". R2C integrates traditional agency knowledge of translational research and development (e.g., WKID Innovation) with resource-based methods motivated by understanding organizational needs (e.g., HCD) [67,68]. By leveraging HCD, R2C is similar to other models proposed for co-production [69] that rely on iterative feedback from users, but it does provide more top–down complex systems analysis to overcome HCD limitations [19–21]. While translational ecology [70] links ecological knowledge to decision making for use-inspired research [66] and real world outcomes [71], R2C uses solutions-oriented science to co-develop technologies providing analytics of value. While co-production models [72,73] and translational ecology [70,71,74] offer mechanisms to build cross-sector partnerships with technology users, they focus on an end-user uptake of information rather than the potential markets to sustain solutions. Translational ecology and co-production models do, however, acknowledge the importance of building workforce with skills beyond academia [74].

While there is a long history of research-to-market pathways for engineering [75], medicine and pharmaceuticals [76], computer science [77], and biology [78], there is a gap for ecology and environmental sciences. The incredible wealth of environmental data from satellite sensors, social media platforms, government records, and other data sources offer remarkable opportunities for market-driven solutions to complex environmental challenges. For example, carbon markets offer a means for offsetting fossil fuel emissions by sequestering carbon [79] or reparative finance for water security [80]. Some companies are starting to use data to verify and validate credits that support improved natural resource management, but there is a need for such analytics to be trustworthy—i.e., transparent, consistent, and secure (Table 2). Co-production between industry and scientific research in academia offers a foundation for developing trustworthy information. Co-production that merges academia with industry requires navigating differences in culture and institutional practices. Through the development of this R2C model, we have learned three key lessons and present some best practices.

First, thought needs to go into how to delineate intellectual property from the start [81] and how a team can co-produce a project together while still maintaining each institutions' rights. A key challenge arises when negotiating contracts and maintaining rights around intellectual property. Legally, "co-production" muddies the water around who owns

what based on contributions [82,83]. A best practice would be to clearly communicate this between all participants and to use multiple modalities for communicating such as the use of diagrams, legal contracts, Gantt charts, and documentation of roles and responsibilities in advance.

Second, agreements on sharing knowledge and data are essential. Such agreements help hold the tension between open science defined as "transparent and accessible knowledge that is shared and developed through collaborative networks" [84] and proprietary information sold as value-added analytics. This requires consideration for establishing data sharing standards [85] accounting for FAIR (Findable, Accessible, Interoperable, and Reproducible) and CARE (Collective Benefit, Authority to Control, Respectful, and Ethical) data principles [86,87] as well as supporting trustworthy AI [50]. Trustworthy AI has 10 considerations. One is the objectivity of data that promotes fairness and mitigates harmful bias. Two is how AI is secured and protected against unauthorized access use, disclosure, disruption, modification or destruction. Three is how AI protects safety and does not cause unacceptable risk. Four is how AI protects privacy. Five is how explainable or transparent an algorithm is. Six is how accurate AI emulates human intelligence. Seven is how reliably AI performs as expected. Eight is how resilient AI is by its adaptive capability. Finally, nine is how accountable AI is for tracing a record of events in the decision-making process.

Third, we recommend developing and implementing a communication plan between institutions to overcome variations in meeting and work cultures. A communication plan [85] that includes a management structure with roles and responsibilities as well as reporting structure within teams [81] can alleviate confusion and enable work management within teams relying on each other for fulfilling their responsibilities of the partnership. Such a plan would alleviate tensions between the different incentive structures and deliverable timelines [88]. For example, a faster cadence of deliverables is required in industry, while academia has a slower turn due to the exploratory nature of the work. Clear technical deliverables, timelines, and inter-team meetings that create "tie-points" between industry and academic workflows enable each team to manage as is culturally appropriate while still collaborating toward the same end purpose [89].

### 4.2. Case Study: Information Gap Analysis and Demonstration Discussion

Recent developments in both research and industry support our findings that dynamic risk and public perception are among the most common information gaps in the wicked wildfire use case. Risk assessments from commercial technologies such as Risk Factor by First Street Foundation and public offerings such as the US Forest Service's WildfireRisk.org or Fuelcast.net are integrating new models for risk [30,90], hazard [37,42,58], exposure [43,91], and vulnerability [92,93]. Combining risk data with evacuation simulations (Figure 7) presents an opportunity for resilience planning with respect to infrastructure planning. Our case study showed that traffic is a predictable consequence of fire evacuations, and it could be mitigated with both planning and technology. Combining long-term fire hazard data with evacuation simulations could inform new road configurations that save lives by preventing entrapment as people move away from high hazard areas (e.g., Appendix A User Stories 3.6 and 3.7). Similarly, social media filtering has become more common for providing information on public perception. Research advancements using social media data show capabilities for geolocating [94–96] and identifying misinformation, deep fakes [97] and unique contributions [11]. The use of these research advancements has yet to be integrated into commercial or public technology offerings.

R2C shows that WKID Innovation and HCD show complementary but not duplicative value for transitioning research to commercialization. WKID Innovation enabled the identification of unique information gaps spanning user types with refined requirements for how to define that information in the context of decisions. WKID Innovation did not provide iterative feedback for how someone might interact with that information in their day-to-day life. HCD provided useful feedback on how users would interact with information in the form of technology (dashboards, apps, etc.), but it neglected to capture the value of social

media filtering to provide information on public perception. This is likely because users do not know to ask for something that does not already exist [20–22]. Furthermore, the feedback from users often resulted in comments that they did not need something that was already similar to a technology they already had but only offered incremental design improvements [19]. R2C demonstrates the need for both a systematic evaluation of information gaps to inform solutions-oriented science and the need for HCD to place the context of that information in a tool that can be commoditized to provide value-added analytics.

## 5. Conclusions

We studied a subset of environmental resilience technology focused on the synthesis and interpretation of data for solutions-oriented science to create value-added analytics that enable society to become more resilient to environmental change. This view of environmental resilience technology addresses the critical need for a more resilient future by shifting away from observing environmental change to providing solutions that serve society.

There is a critical need to leverage domain expertise from research in solutions-oriented science to create value-added analytics that provide sustainable solutions. The research-to-commercialization (R2C) model combines methods for solutions-oriented science (WKID Innovation) and value-added analytics (HCD) while taking advantage of the strengths of academia and industry. R2C facilitates an efficient transfer of knowledge and practice between industry and academic partners. It necessitates academic–industry partnerships that consider how sectors operate with a clear delineation of: (i) intellectual property; (ii) technical deliverables that overcome cultural differences and reward systems; and (iii) a method to both satisfy open science and protect proprietary information and strategy. While we present R2C in a case study for wildfire in the western United States, future work should explore its utility on other environmental resilience topics.

**Supplementary Materials:** The following supporting information can be downloaded at: https://www.mdpi.com/article/10.3390/app131911034/s1, S1: CTM and PTM 20221020. S2: Interview Questions.

**Author Contributions:** Conceptualization, E.N.S., C.G., L.S.D., C.Z., E.D., E.T. and J.K.B.; Data curation, E.N.S., C.G., L.S.D., V.I., C.Z., M.C., J.S. and T.T.; Formal analysis, E.N.S., C.G., L.S.D., C.Z., M.B., L.C. and M.T.; Funding acquisition, E.N.S., E.D., E.T., S.J.M., R.S. and J.K.B.; Investigation, E.N.S., C.G., L.S.D., V.I., C.Z., M.B., M.C., J.S., M.T. and T.T.; Methodology, E.N.S., C.G., L.S.D., C.Z., E.D. and J.K.B.; Project administration, E.N.S., C.G., E.T. and J.K.B.; Resources, E.N.S., S.J.M., R.S. and J.K.B.; Supervision, E.N.S., C.G. and J.K.B.; Validation, C.G., L.S.D., C.Z., M.B., L.C. and M.T.; Visualization, E.N.S., L.S.D., J.S. and T.T.; Writing—original draft, E.N.S., C.G., L.S.D., V.I., M.C., J.S., T.T., E.T. and J.K.B.; Writing—review & editing, E.N.S., C.G., L.S.D., V.I., M.C., J.S., T.T., E.T. and J.K.B. All authors have read and agreed to the published version of the manuscript.

**Funding:** This research was funded by Deloitte Consulting, LLC., through the Climate Innovation Collaboratory.

**Data Availability Statement:** To maintain anonymity of interview participants, we provide synthesized data in the form of the Change Traceability Matrix and Product Traceability Matrix as Supplemental Material to this manuscript.

**Acknowledgments:** We would like to thank Casey Jenson with the Cooperative Institute for Research in Environmental Sciences (CIRES) at the University of Colorado Boulder for helping with Figures 4 and 6. Thank you to Daniel Morton with the Renewable and Sustainable Energy Institute (RASEI) at the University of Colorado Boulder who contributed graphic design for Figure 1. We would also like to thank Jessica Helzer and Emily CoBabe-Amman for their work helping to negotiate the partnership between Deloitte LLC and University of Colorado Boulder.

**Conflicts of Interest:** The funding source (Deloitte Consulting, LLC) for this research precipitated in the design of the research, but the analysis was led by the University of Colorado Boulder who does not receive long-term monetary compensation beyond the contract to do the work for this study and develop the analytics identified by it. Lead author E. Natasha Stavros is the founder of WKID Innovation and has an LLC called WKID Solutions providing education resources on the framework.

**Appendix A. User Stories**

1. Evacuation Route User Stories

   1.1 For protecting lives, property, and the environment, a Local Emergency Responder relies on information such as commute hours with respect to weather, time of day, distance, and fire behavior.

   1.2 For protecting lives, property, and the environment, a Local Emergency Responder relies on information such as Variable Sheriff/Emergency Management response time to communities in need.

   1.3 To effectively communicate with communities (access, e.g., 5G, language, messaging, notifications/alerts, etc.), a Local/Regional Resilience Administrator relies on information of transportation networks (who goes where and how—e.g., public transit, on what ingress/egress routes).

2. Dynamic Risk as a function of hazard, exposure and vulnerability User Stories

   2.1 For protecting lives, property, and the environment, a Local Emergency Responder relies on information such as dynamic risk by parcel based on fuels, weather, and home inspection information.

   2.2 To determine where and when to strategically position resources on the ground at the right location when needed and in response to mutual aid requisitions brokered between the public and local, state, and federal agencies, a State/Regional Emergency Responder relies on information such as dynamic current risk as it relates to anticipated short-term impacts from fire.

   2.3 For Disaster Declaration recommendations sent to the President that determine how much grant dollars are needed for what kind of assistance and for how long to which communities to build capability for state and local level response based on a cost–benefit analysis, a Regional Recovery Administrator relies on information of fire risk.

   2.4 For developing a strategic plan on what mitigation efforts to prioritize based on capability/capacity, infrastructure programs, and social justice that is often vetted with the local community through public engagement exercise and approved by city council/commissioner, a Local Land Use/Land Management/Resilience Planner relies on information of dynamic fire risk as it relates to changing fire hazard (as people cut trees, and structures are built/destroyed as combustible fuels), structural exposure, and structural vulnerability.

   2.5 To develop a wildfire strategy with priority high-risk areas and methods for reducing wildfire risk (fuels management—mechanical, prescribed fire, etc.) decided by rangers in each forest park and local Resilience Offices/County Commissioners and sometimes regionally (most contentious) often communicated and negotiated with the local communities, State/Regional Resilience Administrators rely on information of community Risk updated quarterly that scales from parcel to regional context (e.g., identify highest risk communities locally and regionally).

   2.6 To determine how many staff to hire in support of producing requested analytics by policymakers to assess capacity for meeting legislation mandates, a Resilience Planning Analytics Office relies on information of dynamic risk by parcel (60 m pixel) of assets (structures, power lines, habitat, critical infrastructure, watersheds, etc.) based on fuels, weather and home inspection information.

   2.7 To manage risk/reward trade-offs in a natural perils insurance portfolio by deciding whether or not to take on a risk (e.g., wildfire exposure) and what to charge for that risk based on where it sits within company tolerance for loss as it is written, the property, finance, insurance, reinsurance, and (re-)insurance companies model assets that they want to insure and send it to the underwriter who assesses the premium that can be charged for the risk, and an engineering team may visit the site and assess while offering services like mitigation advice. Underwriting then accepts/rejects risks and may initiate a process with the broker.

Models are run daily on existing reinsured portfolios and monthly on the insured portfolios. This relies on information of dynamic risk (to insured losses) as asset (building) exposure and (building) vulnerability to hazard (not just today, but how it is likely to change).

3. Hazard User Stories

3.1　To determine where and when to strategically position resources (contracted or in-house) on the ground at the right location when needed and in response to mutual aid requests brokered between the public and local, state and federal agencies, a state/regional emergency responder relies on information of dynamic "current" fire "risk" (i.e., hazard) as it relates to changes in fuels, topography and weather.

3.2　To determine where and when to set fuel breaks (e.g., prescribe fire, hand crew, dozer, etc.) during response to active wildfire or in the "shoulder" season, a State/Regional Emergency Responder relies on information of dynamic "current" fire risk as it relates to evolving hazard of fuel condition (stress/moisture, beetles, etc.), type (veg and urban), and accumulation.

3.3　To decide to defend a home or not, a Local/State/Regional Firefighter on the scene relies on information on home building materials.

3.4　To determine how many staff to hire in support of producing requested analytics by policymakers to assess capacity for meeting legislation mandates, a Resilience Planning Analytics Office relies on information of national scale, including rapid, annual updates of vegetation and fuels (updated 3D layers).

3.5　To manage risk/reward trade-offs in a natural perils insurance portfolio by deciding whether or not to take on a risk (e.g., wildfire exposure) and what to charge for that risk based on where it sits within company tolerance for loss as it is written, property, finance, insurance, reinsurance, and (re-)insurance companies model assets that they want to insure and send it to the underwriter who assesses the premium that can be charged for the risk, and an engineering team may visit the site and assess while offering services like mitigation advice. Underwriting then accepts/rejects risks and may initiate a process with the broker. Models are run daily on existing reinsured portfolios and monthly on the insured portfolios. This relies on information of dynamic hazards (not just today, but how it is likely to change).

3.6　To develop a strategic plan on what mitigation efforts to prioritize based on capability/capacity, infrastructure programs, and social justice that is often vetted with the local community through public engagement exercise and approved by city council/commissioner, a Local Land Manager/Land Use/Resilience Planning Administrator relies on information of dynamic fire hazard (as people cut trees, and structures are built/destroyed as combustible fuels).

3.4　To target communications and prepare communities about risk reduction needs and measures (e.g., evacuation routes and planning as well as home hardening), a Local/Regional/National Resilience Administrator relies on information of building locations.

3.8　To develop a wildfire strategy with priority high-risk areas and methods for reducing wildfire risk (fuels management—mechanical, prescribed fire, etc.) decided by rangers in each forest park and local Resilience Offices/County Commissioners and sometimes regionally (most contentious) often communicated and negotiated with the local communities, State/Regional Resilience Administrators rely on information of fuel composition updated quarterly.

4. Vulnerability User Stories

4.1　For Disaster Declarations, Regional Administrators write a recommendation to the President to determine how much grant dollars are needed for what kind of assistance and for how long to which communities to build capability for

state and local-level response based on a cost–benefit analysis. To do this, a Regional Recovery Administrator relies on information of maps of the built environment (structure).

4.2 To build capacity for mitigation through projects that reduce future costs (e.g., debris removal, home hardening, defensible space), State/Regional Recovery Administrators rely on information of projected maps of the built environment (structure).

4.3 To determine where to focus, sheltering resources for both displaced citizens and responders, State/Regional Recovery Administrators rely on information of social vulnerability.

4.4 To develop a strategic plan on what mitigation efforts to prioritize based on capability/capacity, infrastructure programs, and social justice that is often vetted with the local community through public engagement exercise and approved by the city council/commissioner, Local Land Use/Land Management/Resilient Planners rely on information on building vulnerability (ignite-ability) based on factors such as low-income housing, retrofitting, materials, etc.

4.5 To develop a strategic plan on what mitigation efforts to prioritize based on capability/capacity, infrastructure programs, and social justice that is often vetted with the local community through public engagement exercise and approved by city council/commissioner, Local Land Use/Land Management/Resilient Planners rely on information of social equity.

4.6 To manage risk/reward trade-offs in natural perils insurance portfolio by deciding whether or not to take on a risk (e.g., wildfire exposure) and what to charge for that risk based on where it sits within company tolerance for loss as it is written, property, finance, insurance, reinsurance, and (re-)insurance companies model assets that they want to insure and send it to the underwriter who assesses the premium that can be charged for the risk, and an engineering team may visit the site and assess while offering services like mitigation advice. Underwriting then accepts/rejects risks and may initiate a process with the broker. Models are run daily on existing reinsured portfolios and monthly on the insured portfolios. This relies on information of (building) vulnerability.

4.7 To determine whether to defend a home or not, a local emergency response firefighter relies on information such as the Urban Biomass "green biomass" as a Wildland Urban Interface (WUI) layer.

4.8 To support evacuation planning, a Local Resilience Administrator relies on information of social equity.

5. Exposure User Stories

5.1 To determine building capacity for mitigation through projects that reduce future costs (e.g., debris removal, home hardening, defensible space), a State/Regional Recovery Administrator relies on information of projections of built environment (structure) maps.

5.2 To manage risk/reward trade-offs in a natural perils insurance portfolio by deciding whether or not to take on a risk (e.g., wildfire exposure) and what to charge for that risk based on where it sits within company tolerance for loss as it is written, property, finance, insurance, reinsurance, and (re-)insurance companies model assets that they want to insure and send it to the underwriter who assesses the premium that can be charged for the risk, and an engineering team may visit the site and assess while offering services like mitigation advice. Underwriting then accepts/rejects risks and may initiate a process with the broker. Models are run daily on existing reinsured portfolios and monthly on the insured portfolios. This relies on information of dynamic asset (building) exposure.

5.3 To develop a strategic plan on what mitigation efforts to prioritize based on capability/capacity, infrastructure programs, and social justice that is often vetted with the local community through public engagement exercise and approved

by city council/commissioner, a Local Land Manager/Land Use/Resilience Planning Administrator relies on information of dynamic structural exposure.

5.4　To develop a strategic plan on what mitigation efforts to prioritize based on capability/capacity, infrastructure programs, and social justice that is often vetted with the local community through public engagement exercise and approved by city council/commissioner, Local Land Use/Land Management/Resilient Planners rely on information of social equity.

5.5　To manage risk/reward trade-offs in natural perils insurance portfolio by deciding whether or not to take on a risk (e.g., wildfire exposure) and what to charge for that risk based on where it sits within company tolerance for loss as it is written, property, finance, insurance, reinsurance, and (re-)insurance companies model assets that they want to insure and send it to the underwriter who assesses the premium that can be charged for the risk, and an engineering team may visit the site and assess while offering services like mitigation advice. Underwriting then accepts/rejects risks and may initiate a process with the broker. Models are run daily on existing reinsured portfolios and monthly on the insured portfolios. This relies on information of asset locations now and in the future.

6.　Social Media Influence User Stories

6.1　To influence communication strategy for effective communications with communities (access, e.g., 5G, language, messaging, notifications/alerts, etc.), a Local/State/Regional Resilience Administrator relies on information of number of likes and impressions of messaging.

7.　Misinformation User Stories

7.1　To decide how, when and what vetted, validated information (on community needs and situational awareness) to disseminate to the public in a timely manner and where to obtain the information, a Local/Regional Public Information Officer needs to identify point sources of misinformation and misinformation itself.

7.2　To develop a strategic plan on what mitigation efforts to prioritize based on capability/capacity, infrastructure programs, and social justice that is often vetted with the local community through public engagement exercise and approved by city council/commissioner, a Local/Regional Resilience Administrator relies on information of public perception of risk and mitigation efforts with filtered misinformation.

8　Filtered Communications User Stories

8.1　To prioritize the 9-1-1 emergency response dispatch of consolidated resource requests (reducing calls to the right number of resource needs rather than resources/caller who may call about the same event) to the right local agency, a Local Emergency Responder relies on situational information (weapons, threats, etc.).

8.2　To decide when people can return based on hazards and access to utilities (water and power), a Regional Recovery Administrator relies on information on who is evacuating and not evacuating in real time.

8.3　To decide where to focus on sheltering resources for both displaced citizens and responders, a Regional Recovery Administrator relies on information of who needs resources (filtered by social media).

8.4　To decide if resources spent helping on the ground are less than they would receive in consulting on recovery, a Regional Recovery consulting company relies on validated, geolocated information from reliable sources on damages (e.g., downed power lines).

8.5　To decide what agency information to share publicly based on what the public needs to know to reduce the number of duplicate calls on the same incident, a Local Public Information Officer relies on information of evacuations (plans and crowdsourced feedback on available/limited resources and access).

8.6    To decide how, when and what vetted, validated information (on community needs and situational awareness) to disseminate to the public in a timely manner and where to obtain the information, a Regional Public Information Officer uses information to identify the mavens (local media influencers).

8.7    To build public-facing relationships around a cohesive, collaborative strategy across political boundaries for incident response, Incident Command approves staff of the National Incident Management Office (NIMO) as part of the USFS to use validated information on the local event with images, location, timestamps, and information on who took it.

8.8    To determine how and when to pay out on a claim and how to reorganize capital to handle catastrophic events, a (re-)insurance company in national/international resilience planning relies on information of claim validation in the form of geolocation and photos.

9.    Public Perception User Stories

9.1    To protect lives, property, and the environment through response, prevention, and education made locally across departments, Local Emergency Responders coordinated across jurisdictional boundaries ("mutual aid) by the State (e.g., CAL FIRE) with federal resources allocated by Geographic Area Coordination Centers (GACC) rely on information of community perceptions of risk based on fire history and awareness.

9.2    To protect lives, property, and the environment through response, prevention, and education made locally across departments, Local Emergency Responders coordinated across jurisdictional boundaries ("mutual aid) by the State (e.g., CAL FIRE) with federal resources allocated by Geographic Area Coordination Centers (GACC) rely on information of public perception with respect to rumor control of misinformation and ability to turn information into intelligence.

9.3    To develop resilience plans coordinated with each community locally based on watersheds on for planning evacuation routes, infrastructure improvements, where to conduct fuel hazard reductions, and which homes to defend during active response, Local Emergency Responders rely on information of public perception with respect to rumor control of misinformation about resilience measures (e.g., prescribed fire).

9.4    To prioritize the 9-1-1 emergency response dispatch of consolidated resource requests (reducing calls to the right number of resource needs, rather than resources/caller who may call about the same event) to the right local agency, Local Emergency Responders rely on information of public perception of the event.

9.5    To set strategic priorities of how to use limited staff to be successful and where to prioritize investments to reduce risks (e.g., construction tailored to threats) in preparation for an upcoming wildfire season, the grant management of State/Regional Emergency Management rely on information of public perception about prioritization of protecting assets based on variable value systems.

9.6    To decide where and when to strategically position resources (contracted or in-house) on the ground at the right location when needed and in response to mutual aid requests brokered between the public and local, state and federal agencies, State and Regional Emergency Managers rely on information public perception about prioritization of protecting assets based on variable value systems (e.g., timber vs. homes).

9.7    To decide whether to defend a home or not, a local/state/regional firefighter relies on information of public perception about prioritization of protecting assets based on variable value systems (e.g., timber vs. homes).

9.8    To decide where, when and what kind of fuel breaks to allocate (prescribe a fire, hand crew, dozer, goats, etc.) during response to active wildfire or in the "shoulder season", local/state/regional firefighters rely on information of public

perception of fuel treatments and geotagged photos of what is happening; written text needs to be verified in real time (trusted vs. not trusted).

9.9  To decide when to alert and warn people of risk and how to educate the public to take mitigation action, a Local Resilience Administrator relies on information of public perception of events in real time as they happen and that is reliable from trusted sources and accurate with photos and geotagging.

9.10  To develop a strategic plan on what mitigation efforts to prioritize based on capability/capacity, infrastructure programs, and social justice that is often vetted with the local community through public engagement exercise and approved by the city council/commissioner, a Local/Regional Resilience Administrator relies on information of public perception of risk and mitigation efforts with filtered misinformation.

9.11  To decide how to transition from strategic planning to implementation based on priorities of the local community identified by and ranked by the city Chief Resilience Officer, a Local/Regional Resilience Administrator relies on information of public perception of risk and mitigation efforts with filtered misinformation.

9.12  To communicate and prepare communities about risk reduction needs and measures (e.g., evacuation routes and planning as well as home hardening), a Local/Regional Resilience Administrator relies on information of public perception and understanding of fire risk and preparedness as well as public sentiment to determine messaging to communities of fire expectations.

9.13  To influence communication strategy for effective communications with communities (access, e.g., 5G, language, messaging, notifications/alerts, etc.), a Local/State/Regional Resilience Administrator relies on information of the number of public sentiments to determine buy-in of assets being protected.

9.14  To decide how, when and what vetted, validated information (on community needs and situational awareness) to disseminate to the public in a timely manner and where to obtain the information from within the constraints and scope directed by an Incident Commander, a Local/Regional Public Information Officer relies on information of public sentiment of the event.

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
