# Peer review of "Environmental Resilience Technology: Sustainable Solutions Using Value-Added Analytics in a Changing World"

_applsci, doi:10.3390/app131911034_

Round 1
Reviewer 1 Report
Dear Authors,
The content of your paper is quite interesting: your analysis of what went wrong in the local community, by looking at twitter messages as well as by well-structured interviews. Tables 1 and 2 are very useful in practice.
Your WKID framework is interesting, and seems to be essential for the analysis.
However, the text of your paper is awful: in structure, vocabulary as well as grammar. You claim in your text much more than you deliver, your introduction is not focused on the case that you describe. You seem to make your system more important than that it is. As a reader I struggled a lot with what I perceived as ‘hot air’ and ‘mumbo jumbo’ all over the place.
Specific remarks:
(1) the title does not cover the content, since the title is far too broad. I would suggest to change the title to something like: ”Analyzing complex socio-economic systems by means of the WKID framework: the Colorado Marshall Fire”
(2) your text on the WKID framework in Introduction section 1.1 is long, hard to read, and has a lot of mumbo jumbo. Your text at eartlab.colorado.edu/blog/wicked-problems-need-wkid-innovation is much better: only after reading this text I understood your system
(3) you seem to do marketing for Deloitte in section 1.2. You claim “In this paper we develop and apply a Research to Commercialization (R2C)”, however, (a) it is not clear to the reader what is so special of R2C in addition to the WKID framework (b) the paper is not at all clear about the development of value added systems: section 4 and 5 describe the way commercial systems might be developed in general, but nothing about specific systems that have been developed and introduced in this case (the final phases of your WKID). In other words: it is not clear why R2C has been introduced as a separate system, and what has been achieved (other than advice on legal contracts on IP issues): you seem to talk only about ‘opportunities’ (page19).
My conclusion:
(a) the content of your paper is interesting enough to be published, however the text is hard to read an even sometimes irritating.
(b) It is the decision of the editor and you to publish this paper as it is, or improve it
Text is hot air, mumbo jumbo, all over the place. Authors claim more than that they deliver, and use unnecessary (self invented) abbreviations. The introduction should be much shorter and better focused. The Conclusion is disappointing.
Author Response
Please see the attached document for a point-by-point response.

Reviewer 2 Report
The manuscript is well made, scientifically exact, sophisticatedly processed, I do not find any flaws that would lead to the need to change or rework something. My comments are only formal, they are "cosmetic" in nature, I would have expected more cited MDPI sources and a more detailed description of not only the conceptual framework, but also research methods and techniques. On the other hand, I appreciate the reference to a research tool, based on which one can indirectly form an opinion about the philosophy and specific techniques of inquiry. Therefore, in my opinion, there is no need to revise the article yet. I congratulate the authors on an excellent scientific work.
Author Response

(The authors gave the same response as above.)

Reviewer 3 Report
Dear Author,
The paper is well-organized and well-prepared. Few arrangements will make the paper have higher quality. Please check the comments below:
1. Line 9-13, the affiliationnumber “4” is missing. Please check.
2. Line 15-17, these affiliations are already given above.
3. It would be better to add a new title as Data Sets that present information about the whole data sets used in the study.
4. Line 251, please remove the duplication of “Research to”.
5. Figure captions should be given under the Figures. Please check the whole text and correct if necessary.
6. Line 577, concerning the Figure 8, I could not find any information how you generated the change map between 1990 and 2019, and then how you generated the simulation (future form) of changes between 2020-2060.
The paper is well-written in English. Minor editing of English language required.
Author Response

(The authors gave the same response as above.)

Reviewer 4 Report
Comments
I have read the manuscript titled "Environmental Resilience Technology: Sustainable Solutions Using Value-Added Analytics in a Changing World" submitted to Applied Sciences. This paper has some research significance and the workload is large. But I recommend Accept this manuscript after major revision. Here I have four comments on how to make manuscript better:
1. For Figure 5, please check the “%” on the vertical coordinate and it’s title part. Are there any unnecessary issues?
2. For Figure 8, please use the same color scale bar so as to discover the change part easily.
3. Generally, the layout of a diagram starts with the diagram and ends with its title. Please carefully check the figures.
4. Please use the third person to discuss the whole paper.
Author Response

(The authors gave the same response as above.)

Round 2
Reviewer 1 Report
Dear Authors,
Unfortunately your story is hardly improved.
That is a pity: (1) with such a vague title you will hardly attract readers (2) with such an abstract and introduction, most of the readers that you have attracted, will stop reading before they come to the interesting stuff in the middle of your paper.
Conclusion: a missed opportunity, but I will accept the paper in its present form.